# Could traces of fluoroquinolones in food induce ciprofloxacin resistance in *Escherichia coli* and *Klebsiella pneumoniae*? An *in vivo* study in *Galleria mellonella* with important implications for maximum residue limits in food

Zina Gestels,[1] Yuliia Baranchyk,[1,2] Dorien Van den Bossche,[3] Jolein Laumen,[1] Said Abdellati,[3] Basil Britto Xavier,[1,4] Sheeba Santhini Manoharan-Basil,[1] Chris Kenyon[1,5]

**ABSTRACT**  We hypothesized that the residual concentrations of fluoroquinolones allowed in food (acceptable daily intake—ADIs) could select for ciprofloxacin resistance in our resident microbiota. We developed models of chronic *Escherichia coli* and *Klebsiella pneumoniae* infection in *Galleria mellonella* larvae and exposed them to ADI doses of ciprofloxacin via single dosing and daily dosing regimens. The emergence of ciprofloxacin resistance was assessed via isolation of the target bacteria in selective agar plates. Exposure to as low as one-tenth of the ADI dose of the single and daily dosing regimens of ciprofloxacin resulted in the selection of ciprofloxacin resistance in *K. pneumoniae* but not *E. coli*. This resistance was associated with cross-resistance to doxycycline and ceftriaxone. Whole genome sequencing revealed inactivating mutations in the transcription repressors, *ramR* and *rrf2*, as well as mutations in *gyrA* and *gyrB*. We found that ciprofloxacin doses 10-fold lower than those classified as acceptable for daily intake could induce resistance to ciprofloxacin in *K. pneumoniae*. These results suggest that it would be prudent to include the induction of antimicrobial resistance as a significant criterion for determining ADIs and the associated maximum residue limits in food.

**IMPORTANCE**  This study found that the concentrations of ciprofloxacin/enrofloxacin allowed in food can induce *de novo* ciprofloxacin resistance in *Klebsiella pneumoniae*. This suggests that it would be prudent to reconsider the criteria used to determine "safe" upper concentration limits in food.

**KEYWORDS**  *Klebsiella pneumoniae*, *Escherichia coli*, antimicrobial consumption, AMR, resistance, quinolone, minimum selection concentration, MSC

Are the concentrations of antimicrobials allowed in the food we eat able to select antimicrobial resistance (AMR) in our resident bacteria? We do not know the answer, but a growing body of evidence suggests this may be possible (1). Over 10 years ago, Gullberg et al. provided evidence that antimicrobial concentrations up to 230-fold lower than the minimal inhibitory concentration (MIC) are capable of selecting resistant versus susceptible strains of *Escherichia coli* and *Salmonella enterica* spp. (2, 3). Gullberg et al. defined the minimum concentration of an antimicrobial that is able to select for antimicrobial resistance as the minimum selective concentration (MSC) (2, 4). There are two types of MSC. The first is the minimum concentration of an antimicrobial at which one can induce *de novo* resistance ($MSC_{denovo}$). The second is the lowest antimicrobial concentration that selects for a resistant compared to a susceptible strain ($MSC_{select}$) (2,

Address correspondence to Chris Kenyon, ckenyon@itg.be.

Sheeba Santhini Manoharan-Basil and Chris Kenyon contributed equally to this article. The order of the first authors was determined by discussion amongst all the authors.

The authors declare no conflict of interest.

5). Gullberg et al. found the *E. coli* ciprofloxacin $MSC_{select}$ to be 230-fold lower than the MIC and the $MSC_{denovo}$ to be at least 10-fold lower than the MIC. They did not assess if lower concentrations could select for *de novo* resistance (4). More recent experiments have established that ciprofloxacin concentrations as low as 4 ng/L, or 1/1,000th of the MIC, can induce resistance in *Neisseria gonorrhoeae* (6). A systematic review of 62 studies found that subinhibitory concentrations of fluoroquinolones were able to select for increases in MIC for a wide range of Gram-positive and -negative organisms (7). The mechanisms responsible included both target mutations and enhanced expression and activity of efflux pumps (7). No study that we are aware of has assessed MSCs *in vivo*.

The published ciprofloxacin MSCs of *E. coli* and *N. gonorrhoeae* are considerably lower than the concentration of quinolones in meat, water, and environmental samples from a number of locations. Studies from East Asia, for example, have found the mean concentration of ciprofloxacin in eggs and edible fish samples to be 16.8 and 331.7 µg/kg, respectively (8–10). Furthermore, the consumption of foods with high quinolone concentrations has been found to be associated with high urinary and fecal concentrations of quinolones in humans (11–14). For example, a study from South Korea found that high concentrations of enrofloxacin and ciprofloxacin in the urine of the general population were strongly associated with the consumption of beef, chicken, and dairy products (11). Likewise, a large study of the general population in three regions of China detected ciprofloxacin, enrofloxacin, and ofloxacin above the level of detection in the feces of 67%, 30%, and 57% of individuals, respectively (14). The authors attributed the high median concentration of quinolones (median 20 µg/kg) to the ingestion of veterinary antimicrobials in food (15). Reducing the consumption of these foodstuffs has also been found to result in a reduction of urinary quinolone concentrations (15). A number of studies have found the consumption of animal products to be a risk factor for AMR. The Rotterdam Study, for example, was a prospective cohort study that found that ciprofloxacin resistance in community-acquired urinary tract infections was associated with a high intake of pork and chicken (16).

Ecological studies from Europe have also found positive associations between quinolone resistance in *E. coli* bloodstream infections in humans and the total consumption of quinolones in animals (17). Importantly, studies have also shown that reducing antimicrobial use in food-producing animals can result in reduced AMR in humans (18).

In a recent survey of the general population in Belgium, we found that individuals' commensal *Neisseria* species had high ciprofloxacin, azithromycin, and ceftriaxone MICs compared to historical cohorts. This was despite the fact that none of the individuals had knowingly ingested antibiotics in the prior 6 months (19–21). Some of the participants with resistant commensals had not ingested antibiotics for over 30 years (20). These findings led us to hypothesize that antimicrobial exposure from other sources, such as residues in food, may play a role.

The European Medicines Agency (EMA) calculates the acceptable daily intake (ADI) of a medicinal compound based on studies evaluating thresholds for different types of toxicity (22–27). For example, if the ADI for a compound is 100 µg/person for microbiological toxicity and 200 µg/person for cellular toxicity, then the EMA uses the lower ADI of 100 µg/person for that compound. For the quinolones, all the EMA ADIs are determined based on microbiological toxicity (22, 28, 29). These are established by evaluating the MICs for common human bacterial commensals/pathobionts such as *E. coli* and calculating estimated dose exposures in the human colon (22–24, 27, 30). The induction of or selection for AMR is not considered (27). The ADIs are then used to calculate maximum residue limits (MRLs)—the maximum concentration of the compound allowed in food products based on the average consumption patterns of those food products (27, 31).

The most recent EMA reports concluded that the acceptable daily intake of enrofloxacin is 6.2 µg/kg (22, 32). We hypothesized that this dose could induce resistance *in vivo*. We tested this hypothesis in a *Galleria mellonella* model of chronic *E. coli* and *K.*

*pneumoniae* infection treated with ADI equivalent concentrations of ciprofloxacin. We chose to use ciprofloxacin rather than enrofloxacin for a number of reasons. First, most enrofloxacin is metabolized to ciprofloxacin. This results in the ciprofloxacin concentration exceeding enrofloxacin concentrations post-ingestion of enrofloxacin in food-producing animals such as cows (33). Second, numerous surveys of quinolone detection in human urine samples from the general population in East Asian countries have found that ciprofloxacin was detected in a higher proportion of individuals and at higher concentrations than enrofloxacin (11–14, 34, 35). Consumption of these antimicrobials in food was thought to be the most likely source of these antimicrobials (11–14). Third, the ciprofloxacin and enrofloxacin MICs for *Enterobacteriaceae* are almost identical (36, 37). We assessed two doses of ciprofloxacin—ADI dose (6.2 µg/kg) and 1/10th of this dose.

## MATERIALS AND METHODS

### Bacterial strains and growth conditions

We selected two strains of *K. pneumoniae* and one strain of *E. coli* for the experiment. The main criterion for selecting these strains was a low ciprofloxacin MIC. The two *K. pneumoniae* strains were clinical isolates from our Institute of Tropical Medicine collection, and the *E. coli* strain was a clinical isolate from the ATCC collection. Detailed information on the bacterial strains used in this study is provided in Table 1.

### Preparation of live microbial inocula for infection

The strains of *E. coli* and *K. pneumoniae* were cultured from frozen stocks onto BD Columbia Agar with 5% sheep blood for ≤16 h at 37°C with 5% (vol/vol) $CO_2$. Single colonies were selected and spread onto fresh agar plates, which were incubated at 37°C with 5% (vol/vol) $CO_2$ for 6 h. *E. coli* and *K. pneumoniae* were then inoculated into the hemocoel of the *Galleria mellonella* larvae [10 µL of phosphate-buffered saline (PBS) containing $10^3$ CFU/larva]. This dose of *E. coli* and *K. pneumoniae* was determined based on experiments that established a dose that enabled the recovery of the bacteria up to 5 days post-inoculation without killing the larvae (data not shown).

### Injection of *G. mellonella* larvae

The last larval stage of *Galleria mellonella* (Terramania, Arnhem, NL) was used for the experiments. Only macroscopically healthy, non-discolored larvae of 250–450 mg were selected. The larvae were placed into individual sterile Petri dishes in groups of 10 per Petri dish. The larvae were kept in an incubator at 37°C with a 5% (vol/vol) $CO_2$ atmosphere for the length of the experiments.

The larvae were injected in the last right pro-leg with 10 µL of bacterial suspension, followed 10–20 min later by various doses of ciprofloxacin, using 0.3 mL U-100 insulin syringes (BD Micro-Fine). One syringe and needle were used for 10 larvae in each Petri dish (Fig. 1).

### Dose of ciprofloxacin injected

As noted above, the EMA defines the acceptable daily intake of enrofloxacin as 6.2 µg/kg (22, 32). This translates into a dose of 2.36 ng for 380 mg *G. mellonella* (the average weight of the larvae used in our experiments) (Table S1). We tested two doses of ciprofloxacin—2 ng (ADI dose) and 0.2 ng (0.1 × ADI) per larva.

**TABLE 1** Bacterial strains used in this study

| Organism | Isolate number | Ciprofloxacin MIC (µg/mL) | Clinical origin |
|---|---|---|---|
| *K. pneumoniae* | M14827 | 0.047 | ITM clinical isolate |
| *K. pneumoniae* | M17125 | 0.047 | ITM clinical isolate |
| *E. coli* | ATCC 25922 | 0.023 | *E. coli* Seattle 1946 is a clinical isolate from the ATCC collection |

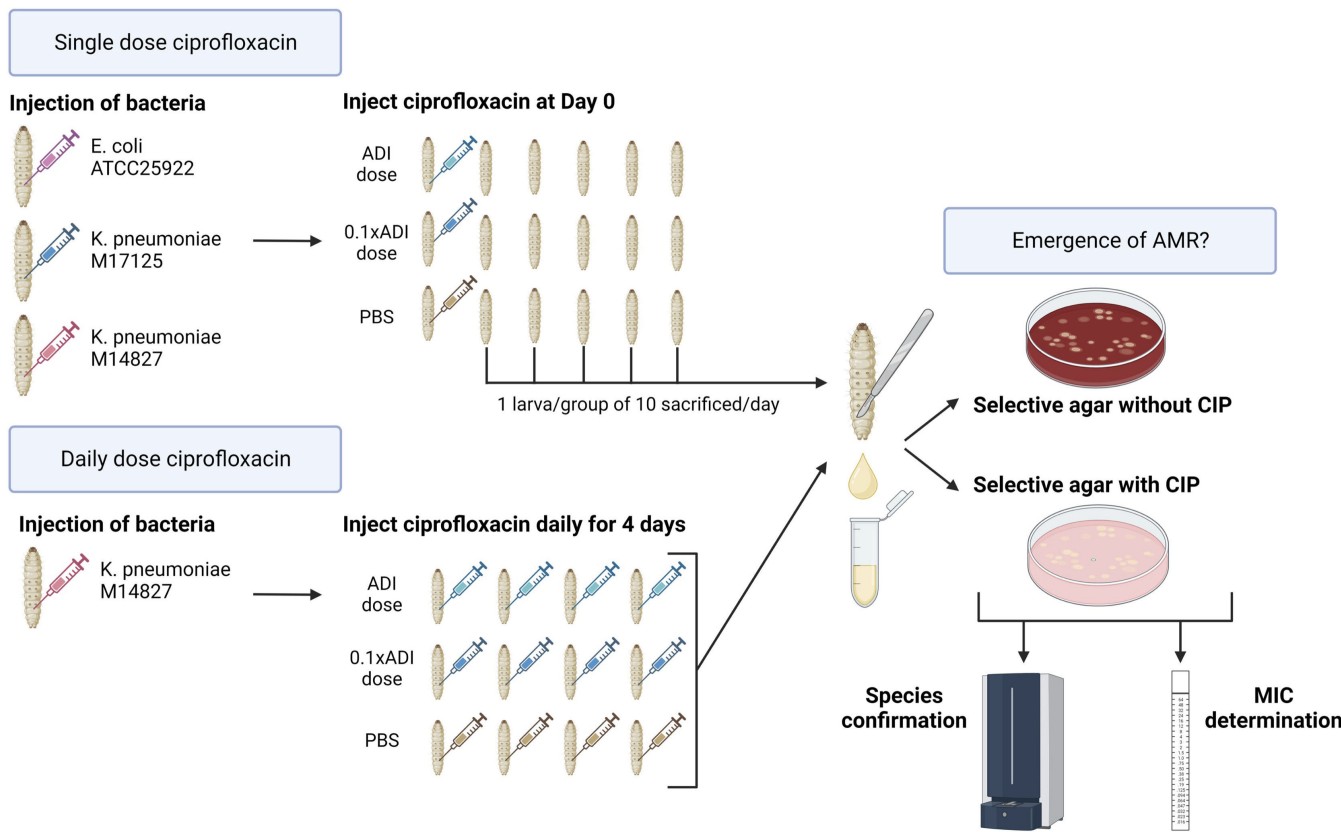

**FIG 1** Schematic overview of study methodology (figure produced with BioRender).

When each experiment was completed, both the surviving and dead *G. mellonella* were kept at −80°C overnight to sedate and kill the surviving larvae. They were then autoclaved at 121°C for 15 min and discarded.

## Retrieval of *E. coli/K. pneumoniae* from *G. mellonella*

At 24 h after the injection of the bacteria and 24-h intervals thereafter, one larva from each group of 10 larvae was randomly selected for the extraction of the hemolymph. This was continued for the duration of the experiments—5 days. The larvae were placed at −80°C for 60 seconds until no movements were observable. They were then placed on a Petri dish, and an incision was made between the two segments closest to the tail of the larva. Hemolymph was then extracted by squeezing the hemolymph into 1.8-mL centrifuge tubes containing 50 µL PBS. The hemolymph from each larva was then vortexed and divided onto selective plates with and without ciprofloxacin: *E. coli*: Chromocult Coliform Agar [CCA; Merck (Darmstadt, Germany)] with 0.125 µg/mL ciprofloxacin and CCA without ciprofloxacin; *K. pneumoniae*: *Klebsiella* ChromoSelect Selective Agar [KA; Merck (Darmstadt, Germany)] with 0.125 µg/mL ciprofloxacin and KA without ciprofloxacin.

The CCA/KA plates were then incubated at 37°C with a 5% (vol/vol) $CO_2$ atmosphere for 24 h, and then, the number of purple-magenta (*K. pneumoniae*) or blue (*E. coli*) colonies was counted. All the blue and purple-magenta colonies growing on the ciprofloxacin plates and a random selection of up to two single colonies from the no-ciprofloxacin plates were selected for identification via matrix-assisted laser desorption/ionization-time-of-flight mass spectrometry (MALDI-TOF MS) and ciprofloxacin MIC determination via Etest (AB bioMerieux, France). The Etests were performed on Mueller–Hinton agar plates incubated for 16–18 h at 37°C with a 5% (vol/vol) $CO_2$ atmosphere. We followed the EUCAST guidelines in defining ciprofloxacin resistance

as ≥0.06 µg/mL for *E. coli* and ≥0.125 µg/mL for *K. pneumoniae* (https://www.eucast.org/mic_and_zone_distributions_and_ecoffs).

## Control groups

For all experiments, a main AMR control group was included that received the same protocol—bacterial inoculation followed by 10 µL/larva of PBS. Each control and experimental group consisted of at least 30 larvae. To ensure that we were not detecting *E. coli* and *K. pneumoniae* from the larval microbiota, we included two further species control groups ($n = 10$, each group) that were inoculated with PBS, and then, on day 5, their hemolymph was extracted and plated onto selective agar for either *E. coli* (Chromocult Coliform Agar) or *K. pneumoniae* (Klebsiella ChromoSelect Selective Agar) as described above.

## MALDI-TOF MS species identification

The isolates were confirmed to be *K. pneumoniae* using MALDI-TOF MS, on a MALDI Biotyper Sirius IVD system using the MBT Compass IVD software and library (Bruker Daltonics, Bremen, Germany). Each bacterial isolate was spread on a polished steel target plate, covered in 1 µL of muric acid, dried, and then covered with 1 µL of α-cyano-4-hydroxycinnamic acid matrix solution. After the reagents had dried, the target plate was loaded. The spectra were acquired in a linear mode in a mass range of 2–20 kDa and then compared to the library. Identification results were classified as reliable or unreliable according to recommended cut-off values of 1.7 and 2 for validated results for the genus and species levels, respectively.

## Single-dose experiments

The single-dose experiments were performed on larvae injected with *K. pneumoniae* ($n = 2$) and *E. coli* ($n = 1$) strains. In these experiments, each group of larvae received a single dose of ciprofloxacin—ADI dose, $0.1 \times$ ADI, or PBS, at the beginning of the experiments.

## Daily-dose and cross-resistance experiments

A single strain of *K. pneumoniae* (M14827) was used for the evaluation of daily-dose ciprofloxacin and cross-resistance. For this experiment, each group of larvae was injected with a daily dose of ciprofloxacin (ADI dose, 0.1 ADI or PBS) for 4 days. To assess for cross-resistance, all isolates grown on the ciprofloxacin-containing plates and confirmed by MALDI-TOF MS to be *K. pneumoniae* had their doxycycline and ceftriaxone MICs determined using Etest in triplicate (AB bioMerieux, France).

## Data analysis

Statistical analyses and data visualization were conducted using GraphPad Prism with Mann–Whitney or ANOVA tests used to compare groups, depending on Gaussian distribution. A *P*-value <0.05 was considered statistically significant.

## Whole genome sequencing

A random selection of 36 *K. pneumoniae* isolates with the highest evolved ciprofloxacin MICs and two controls were selected for whole genome sequencing (Tables S1 and S2). The samples were outsourced to Eurofins, where total DNA was isolated. Library preparation was carried out using a Stranded TruSeq DNA library preparation kit. The sequencing was performed on NextSeq 6000, v2, $2 \times 150$ bp (Illumina Inc., San Diego, CA, USA). For the whole genome sequencing (WGS) analysis, initial quality control of the raw reads was carried out using FastQC (38). The raw reads were trimmed using trimmomatic (v0.39) (Phred score 33 and length of the bases ≥30 bases) (39). The processed raw reads were *de novo*-assembled using Shovill (v1.0.4) (40), which uses SPAdes (v3.14.0) using the following parameters: trim—depth 150—opts—isolate (41).

The quality of the *de novo*-assembled contigs was evaluated using Quast (v5.0.2) and annotated using Prokka (v1.14.6) (42, 43). The quality-controlled reads were mapped to respective reference draft genomes [M14827-2-A (Table S1) and M17125-1A (Table S2)] using CLC Genomics Workbench (v20). Single-nucleotide polymorphisms were identified using the basic variant detection tool implemented in CLC Genomics Workbench (v 20) with parameters 10× minimum read coverage and a minimum frequency of 35% at the variant locus. The raw reads generated are deposited at BioProject ID PRJNA974953.

## RESULTS

### Colonization

The introduced species *E. coli* (*n* = 1) and *K. pneumoniae* (*n* = 2) were recovered upon culturing on a selective agar plate for up to 5 days after inoculation (Fig. S1 and S2) and were verified by MADLI-TOF MS. No *E. coli* or *K. pneumoniae* were cultured from the larvae injected with PBS only.

### Mortality

For each of the three bacterial strains, there was no difference in the mortality rates of the larvae between the ciprofloxacin-treated and control groups (Fig. S3).

### Emergence of AMR

#### Single-dose experiment

The emergence of ciprofloxacin resistance was primarily assessed via counting colonies on the selective agar plates with ciprofloxacin. No colonies were seen on any of the ciprofloxacin plates from the control groups or the *E. coli* group that received ciprofloxacin (Fig. 2). A single colony of *K. pneumoniae* M17125 (ciprofloxacin MIC 0.38 µg/mL) was recovered at day 5 from the ciprofloxacin plates that had received the ADI dose of ciprofloxacin (Fig. 2). For the *K. pneumoniae* M14827 group, a number of resistant colonies emerged in the ADI and 0.1 × ADI dose groups on day 1, which peaked at day 2, and a smaller number of colonies were detected until day 5 for both groups (Fig. 2). The median, interquartile range (IQR) ciprofloxacin MICs of these colonies increased from 0.047 µg/mL (0.047–0.047 µg/mL) to 0.25 µg/mL (0.19–0.38 µg/mL; *P* < 0.0001) and

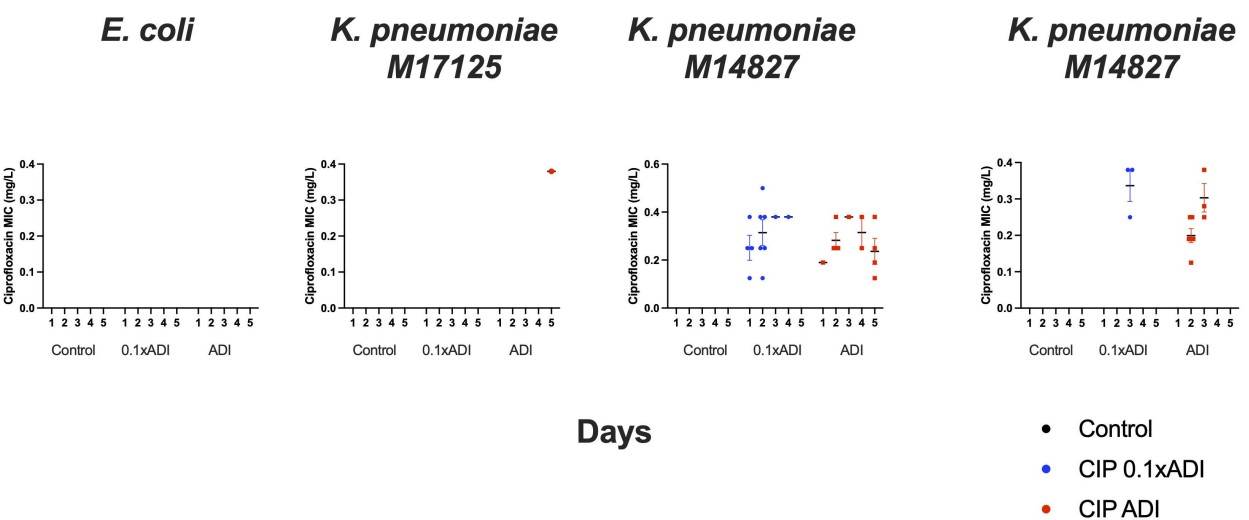

**FIG 2** Ciprofloxacin MICs of bacterial colonies of *E. coli*, *K. pneumoniae* M14827, and *K. pneumoniae* M17125 after exposure to ciprofloxacin (ADI dose, 0.1 × ADI dose, or control/PBS) taken from selective plates with ciprofloxacin (0.125 µg/mL). Mean and SEM shown.

0.31 µg/mL (0.25–0.38 µg/mL; $P < 0.0001$) for the ADI and 0.1 × ADI groups, respectively (Fig. 2). These MICs were five to seven times greater than the baseline MIC, and these MICs were significantly greater than the MICs of the colonies obtained from the control arm at day 5 [median MIC 0.047 µg/mL (IQR 0.047–0.064 µg/mL); $P < 0.0001$ for both comparisons].

We also assessed the ciprofloxacin MICs of two randomly selected colonies per plate from the plates without ciprofloxacin. For *E. coli* and both strains of *K. pneumoniae*, the ciprofloxacin MICs were not higher in the CIP 0.1 × ADI or the CIP ADI than in the control groups (Fig. S4).

### Daily-dose experiment

This experiment was only conducted on *K. pneumoniae* M14827. The receipt of daily ADI doses was associated with an increase in ciprofloxacin MIC on days 2 and 3. The median (IQR) ciprofloxacin MIC increased from 0.047 µg/mL (0.047–0.047 µg/mL) to 0.25 µg/mL (0.19–0.38 µg/mL; $P = 0.0075$) in the ADI group (Fig. 2). In the group receiving 0.1 × ADI daily, the ciprofloxacin MIC increased from 0.047 µg/mL (0.047–0.047 µg/mL) to 0.25 µg/mL (0.19–0.38 µg/mL; $P = 0.0236$) in isolates from day 2, the only day when colonies were visible on the ciprofloxacin plates. These emergent MICs were also greater than the MICs of the *K. pneumoniae* exposed to PBS daily for 5 days (Fig. S5).

### Emergent mutations

Putative emergent mutations were detected in four genes in the groups exposed to ciprofloxacin and not the controls (Table 2). In all the experimental groups of *K. pneumoniae* M14827 exposed to various dosing schemas of ciprofloxacin, mutations emerged in *Rrf2* family transcriptional regulator and *ramR*. These mutations were predominantly frameshift (fs) mutations. In addition, a Gly81Cys mutation in GyrA and a Lys10fs mutation in RamR emerged on day 2 in the daily ADI ciprofloxacin group. On day 2 in this same group, a Pro747Ser mutation emerged in conjunction with a Leu76Gln mutation in Rrf2. No relevant mutations were detected in the single isolate of *K. pneumoniae* M17125 with an elevated ciprofloxacin MIC. A list of all mutations detected is provided in Tables S1 and S2.

### Cross-resistance

The emergent median (IQR) ceftriaxone MICs, 0.125 µg/mL (IQR 0.125–0.38 µg/mL) of *K. pneumoniae* M14827 obtained from the ciprofloxacin plates of the daily-dose experiment, were elevated compared to the parental strains [0.064 µg/mL (IQR 0.064–0.064 µg/mL); $P < 0.001$; Fig. 3]. Likewise, the median (IQR) doxycycline MICs [16 µg/mL (IQR 16–32 µg/mL)] were elevated compared to the parental strains [2 µg/mL (IQR 0.064–0.064 µg/mL); $P < 0.001$; Fig. 3.

### DISCUSSION

In our *G. mellonella* model, we found that the receipt of as low as a single dose of one-tenth of the acceptable daily intake of ciprofloxacin was followed by an up to sevenfold increase in ciprofloxacin MIC. Daily exposure to ciprofloxacin did not result in higher MICs than single-dose exposure but was associated with the emergence of Gly81Cys and Pro747Ser mutations in GyrA and GyrB, respectively. Both of these mutations have been shown experimentally to be causally linked to the emergence of fluoroquinolone resistance in a number of Gram-negative bacteria (44–47).

Mutations in the quinolone resistance determinant region (QRDR) of *gyrA*, *gyrB*, *parC*, and *parE* are important determinants of resistance to fluoroquinolones (48). The three main other resistance mechanisms are (i) increased expression of efflux pump(s), (ii) *qnr* genes that protect the target topoisomerase, and (iii) the presence of aac(60)-lb-cr that acetylates fluoroquinolones (48). Neither *qnr* nor aac(6′)-lb-cr genes were detected in our

**TABLE 2** Emergent mutations and ciprofloxacin MICs of *K. pneumoniae* M14827 following exposure to low-dose ciprofloxacin[b]

| Experimental group | WGS ID | Ciprofloxacin MIC (µg/mL) | Days | Ciprofloxacin dose | Rrf2 (OGBGAAEN_04402) | RamR (OGBGAAEN_00960) | DNA gyrase subunit A | DNA gyrase subunit B |
|---|---|---|---|---|---|---|---|---|
| Single dose | KPCE3-1 | 0.25 | 1 | 0.1 ADI | – | (TA ins) Asp94fs | – | – |
| Single dose | KPCE3-6 | 0.25 | 2 | ADI | – | (GCGT del) Val97fs | – | – |
| Single dose | KPCE3-8 | 0.25 | 2 | 0.1 ADI | (TGCTCCGCCAT ins) Met22fs | – | – | – |
| Single dose | KPCE3-9 | 4 | 2 | ADI | Leu35Arg | – | – | – |
| Single dose | KPCE3-10 | 0.38 | 2 | ADI | Ser26[a] | – | – | – |
| Single dose | KPCE3-13 | 0.25 | 4 | ADI | – | (ACAAAG del) His103_Ala105delinsPro | – | – |
| Single dose | KPCE3-14 | 0.38 | 3 | ADI | (CTCCGCCATTGCTA del) Val16fs | – | – | – |
| Single dose | KPCE3-17 | 0.25 | 5 | ADI | – | Ala40Val | – | – |
| Single dose | KPCE3-19 | 0.3 | 5 | 0.1 ADI | (ACGGT del) Thr129fs | – | – | – |
| Daily dose | Kpdaily1 | 0.25 | 3 | ADI | Ser64Leu | – | – | – |
| Daily dose | Kpdaily3 | 0.38 | 3 | ADI | (CCAGCGCGCTGGC del) Ala131fs | – | – | – |
| Daily dose | Kpdaily4 | 0.25 | 3 | 0.1 ADI | (CGTTGGCGCTGACCA del) Val101_Asn105del | – | – | – |
| Daily dose | Kpdaily5 | 0.38 | 3 | 0.1 ADI | Arg61Ser | – | – | – |
| Daily dose | Kpdaily6 | 0.38 | 3 | 0.1 ADI | Ser92[a] | – | – | – |
| Daily dose | Kpdaily7 | 0.25 | 2 | ADI | – | (-del) Val67fs | – | – |
| Daily dose | Kpdaily9 | 0.19 | 2 | ADI | – | (-del) Lys10fs | Gly81Cys | – |
| Daily dose | Kpdaily10 | 0.19 | 2 | ADI | – | Gln122[a] | – | – |
| Daily dose | Kpdaily11 | 0.19 | 2 | ADI | Leu76Gln | – | – | Pro747Ser |
| Daily dose | Kpdaily12 | 0.25 | 2 | ADI | – | (ATCC ins) Arg107fs | – | – |

[a]Premature stop codon.

[b]ADI – Acceptable Daily Intake; Del – deletion; fs - frame shift; ins – insertion. The table is limited to isolates with elevated ciprofloxacin MICs and that were sequenced and only shows mutations that emerged in RrF2, RamR, DNA gyrases A and B. The baseline ciprofloxacin MIC was 0.047 µg/mL, - Not detected.

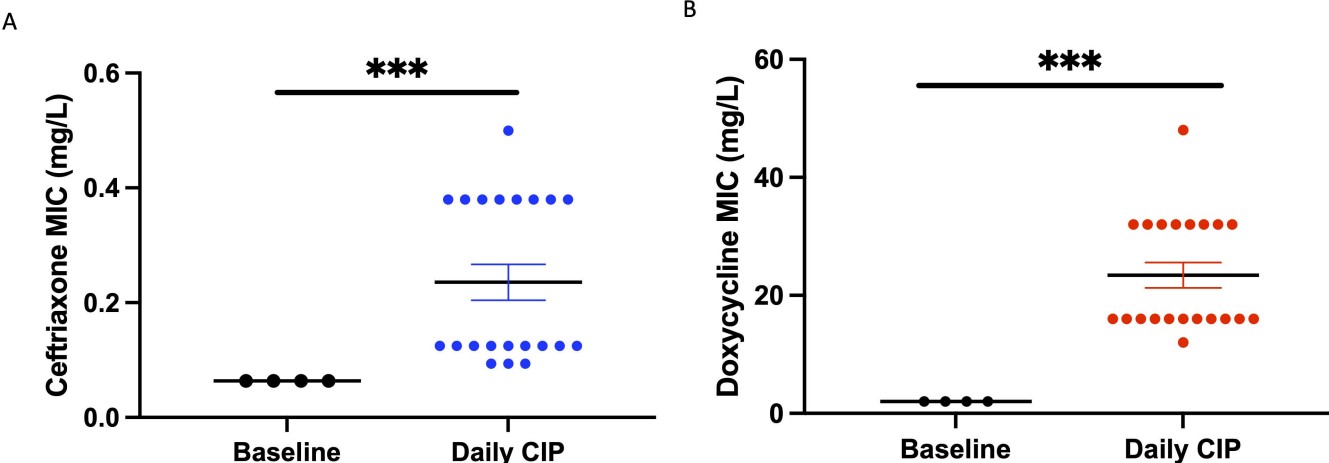

**FIG 3** Selection of cross-resistance. Ceftriaxone and doxycycline MICs of bacterial colonies of *K. pneumoniae* M14827 after daily exposure to ciprofloxacin (ADI and 0.1 × ADI dose combined) versus baseline strain. Mean and SEM shown.

strains. We did, however, detect mutations that may have increased the expression of efflux pumps.

The AcrAB multidrug RND efflux pump has been shown to play an important role in fluoroquinolone resistance and virulence in clinical isolates of *K. pneumoniae* (49–51). This pump is encoded by the *acrRAB* operon, which includes *acrR,* which acts as the AcrAB repressor (49). Studies have established that an increased expression of the pump is caused by mutations in the AcrAB repressors, *acrR* or *ramR*, or overexpression of the transcriptional regulator, *ramA* (52, 53). We found evidence of likely inactivating mutations in *ramR* in eight independent strains of *K. pneumoniae* subjected to low-dose ciprofloxacin. Similar inactivating mutations in *ramR* have been found to be responsible for reduced susceptibility to a range of antimicrobials, including fluoroquinolones, in a high proportion of clinical isolates of *K. pneumoniae* and other Gram-negative bacteria (53–55).

In a similar vein, we found likely inactivating mutations in *rrf2* in seven strains. Rrf2 functions as a repressor of the OqxAB RND efflux pump, and thus, inactivating mutations of *rrf2* have been found to cause increased expression of this pump and, hence, reduced susceptibility to a range of antimicrobials including fluoroquinolones (56–58). Previous studies have found that *rrf2* functions as a repressor of OqxAB and that inactivating mutations in *rrf2* resulted in lower susceptibility to various antimicrobials (57, 59). Other studies have established that fluoroquinolone exposure can induce rrf2 mutations in OqxAB (60).

A previous study found that increased expression of AcrAB was the major mechanism explaining how sub-MIC exposure to enrofloxacin in *Salmonella enterica* serovar *Enteritidis* resulted in reduced susceptibility to fluoroquinolones (61). This study also found that low doses of enrofloxacin resulted in the step-wise acquisition of resistance. The first change was a reduced expression of outer membrane porins, followed by an increased expression of efflux pumps such as AcrAB and then mutations in the QRDR of *gyrA, gyrB,* and *parC*. These results are compatible with our finding that only the *ramR* and *rrf2* mutations were detected in the single-dose experiments, whereas both the *ramR/rrf2* mutations and the *gyrA/gyrB* mutations were found following daily ciprofloxacin exposure.

Other studies of sub-MIC exposure to ciprofloxacin and ceftriaxone in *E. coli* found that low-dose antimicrobials selected for resistant isolates but that the elevated MICs could not typically be explained by known resistance mechanisms (15, 49). In both cases, novel mutations that were thought to explain the increased MICs were detected. Other studies have found that transient mutations in ribosomal proteins can act as stepping

stones to higher-level macrolide and fluoroquinolone resistance (22, 50). Further studies are required to more comprehensively map the molecular pathways to fluoroquinolone resistance following low-dose ciprofloxacin exposure over a longer period. It would be useful if these studies could include other pathways through which low-dose fluoroquinolones could select for AMR. Low-dose fluoroquinolones have been found to activate the SOS response due to DNA damage caused by intereference with DNA gyrase (62, 63). This could lead to enhanced horizontal gene transfer via plasmids, which could in turn induce AMR (62, 64). In addition, ciprofloxacin has been shown to induce persister cells via the SOS response (63). Other studies have found that these persister cells may be more likely to acquire AMR (65).

In addition to only evaluating one antimicrobial in two bacterial species, this study has a number of other limitations. Our model was based on establishing a chronic hemolymph infection in *G. mellonella* and establishing if low doses of an antimicrobial can induce AMR. A more relevant experiment would be to assess if these low doses ingested orally could induce AMR in bacteria in humans or other mammals. Although a range of studies have found that *G. mellonella* infection models (including *K. pneumoniae* and *E. coli*) closely replicate key features of microbial interactions with humans, we have not established if this is the case for the ascertainment of minimum selection concentrations (66, 67). We have also only assessed the effect of low antibiotic doses on the emergence of *de novo* resistance and not on the enrichment of preexisting resistant strains or the spread of mobile genetic elements that confer AMR (2, 6). Furthermore, we did not evaluate if low doses of different classes of antimicrobials, heavy metals, or biocides could act synergistically to generate AMR (68). It is not uncommon for individuals to consume more than one meat or dairy product per day. It is, thus, possible to be exposed to low doses of resistogenic compounds such as fluoroquinolones, macrolides, and heavy metals from three different products (68, 69). It would be prudent for future studies to evaluate if these low doses could interact to induce and select for AMR as has been established *in vitro* (69). We only considered the effect of daily doses of ciprofloxacin for up to 5 days. Future studies could ascertain if longer-term exposure to low doses of antimicrobials could have a greater effect on the emergence of AMR. The stability of the emergent mutations was not evaluated. Previous studies have, however, established that at least some of these mutations are stable. One study, for example, found that the Gly81Cys mutation in GyrB was positively selected in *E. coli*, with 100% maintenance rates in the populations for at least 25 passages (44). Importantly, we did not attempt to establish experimentally if the mutations we detected are causally related to the changes in MICs. Previous studies have, however, established that inactivating mutations in *ramR* and *rrf2* are causally linked to elevations in antimicrobial MICs (53, 57).

Despite these weaknesses, the study is the first of its kind to assess if low doses of antimicrobials can select for AMR *in vivo*. Its positive finding suggests the need for equivalent studies in mammals. If such studies confirm this effect, then antimicrobials ingested in food, water, and beverages could help to explain some of the unresolved determinants of AMR. A number of studies have concluded that the known quantity of antimicrobials ingested by humans does not appear to be able to explain all the observed AMR. For example, certain countries in East Asia are experiencing syndemics of fluoroquinolone resistance in multiple commensal and pathogenic bacteria despite lower fluoroquinolone consumption than less affected countries in other regions (6, 70, 71). The intensive use of fluoroquinolones in food-animal production may contribute to these syndemics as evidenced by country-level studies which, have found positive associations between the intensity of fluoroquinolone and macrolide use for food-producing animals and the prevalence of fluoroquinolone/macrolide resistance in a range of human pathogens (6, 72, 73).

Although we only tested one strain of *E. coli* and two of *K. pneumoniae*, our results appeared to be strain- and species-specific. Almost all the colonies resistant to ciprofloxacin emerged from a single strain of *K. pneumoniae*. This is not too unexpected, as previous experiments assessing the minimum selection concentrations

for fluoroquinolones and other antimicrobials have found large differences between bacterial species and strains (2, 4). Both *E. coli* and *K. pneumoniae* have been proposed as sensitive indicators of excess antimicrobial exposure in various environmental and clinical settings (68, 74). For both species, there is now clear evidence that certain strains are more susceptible to the acquisition and spread of AMR (74). For example, the ST131 H30-Rx strain of *E. coli* has been strongly linked to the spread of the ESBL enzyme, CTX-M15, as well as fluoroquinolone and β-lactam resistance (75). In a similar vein, the clonal group 258 of *K. pneumoniae* has been shown to be responsible for most hospital-acquired infections of carbapenem-resistant *K. pneumoniae* (76). In addition, *Klebsiella* spp. have been shown to play a critical role in the genesis and spread of AMR genes between environmental and human microbial populations (77). The ATCC 25922 strain of *E. coli* we used belongs to phylogenetic group A of *E. coli*, which has been found to be less prone to the emergence of AMR (78). Future experiments may consider using *E. coli* strains from the phylogroups such as B2, which are more associated with the emergence of AMR (78, 79).

Our results suggest that it would be prudent to include the induction/selection of AMR as an important criterion for determining ADIs and MRLs. The *G. mellonella* model offers a useful way to test a large number of bug–drug combinations, but mouse and human models will be required to validate the findings.

## ACKNOWLEDGMENTS

Yuliia Baranchyk was registered in the EMJMD LIVE (Erasmus + Mundus Joint Master Degree Leading International Vaccinology Education), co-funded by the EACEA (Education, Audiovisual, and Culture Executive Agency, award 2018-1484) of the European Commission, and received a scholarship from the EACEA.

This study was supported by a SOFI-B grant, for the PReSTIP Project.

C.K., J.L., S.B., S.A., B.B., I.D.B., D.V.B., and Z.G. conceptualized the study. C.K., J.L., S.A., Y.B., and Z.G. conducted the MSC experiments; B.B. and S.B. were responsible for the bioinformatic analyses and C.K. for the statistical analyses. All authors read and approved the final draft.

## AUTHOR AFFILIATIONS

[1]STI Unit, Department of Clinical Sciences, Institute of Tropical Medicine, Antwerp, Belgium
[2]UnivLyon, Université Claude Bernard Lyon 1, Villeurbanne, France
[3]Clinical and Reference Laboratory, Department of Clinical Sciences, Institute of Tropical Medicine, Antwerp, Belgium
[4]Hospital Outbreak Support Team—HOST, Ziekenhuis Netwerk Antwerpen Middelheim, Antwerp, Belgium
[5]Division of Infectious Diseases and HIV Medicine, University of Cape Town, Cape Town, South Africa

## AUTHOR ORCIDs

Sheeba Santhini Manoharan-Basil http://orcid.org/0000-0001-8421-5137
Chris Kenyon http://orcid.org/0000-0002-2557-8998

## AUTHOR CONTRIBUTIONS

Zina Gestels, Conceptualization, Investigation, Methodology, Writing – review and editing | Yuliia Baranchyk, Conceptualization, Methodology, Writing – review and editing | Dorien Van den Bossche, Conceptualization, Methodology, Writing – review and editing | Jolein Laumen, Conceptualization, Investigation, Methodology, Writing – review and editing | Said Abdellati, Conceptualization, Methodology, Writing – review and editing | Basil Britto Xavier, Methodology, Supervision, Writing – review and editing | Sheeba

Santhini Manoharan-Basil, Conceptualization, Methodology, Writing – original draft, Writing – review and editing.

## ADDITIONAL FILES

The following material is available online.

### Supplemental Material

**Supplementary material (Spectrum03595-23-s0001.docx).** Fig. S1 to S6; Tables S1 and S2.

### Open Peer Review

**PEER REVIEW HISTORY (review-history.pdf).** An accounting of the reviewer comments and feedback.

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
