## [Reviewer comments · Microbiology Spectrum]

Microbiology Spectrum

Could traces of fluoroquinolones in food induce ciprofloxacin resistance in *Escherichia coli* and *Klebsiella pneumoniae*? An in vivo study in *Galleria mellonella* with important implications for maximum residue limits in food

Zina Gestels, Yuliia Baranchyk, Dorian Van den Bossche, Jolein Laumen, Said Abdellati, Basil Britto Xavier, Sheeba Manoharan-Basil, and Christopher Kenyon

Corresponding Author(s): Christopher Kenyon, Instituut voor Tropische Geneeskunde

Review Timeline:

Submission Date:	October 9, 2023
Editorial Decision:	February 22, 2024
Revision Received:	March 5, 2024
Accepted:	April 13, 2024

Editor: Sadjia Bekal

Reviewer(s): Disclosure of reviewer identity is with reference to reviewer comments included in decision letter(s). The following individuals involved in review of your submission have agreed to reveal their identity: Mustafa SADEK (Reviewer #3)

Transaction Report:

DOI: <https://doi.org/10.1128/spectrum.03595-23>

Re: Spectrum03595-23 (Could traces of antibiotics in food induce antimicrobial resistance in *Escherichia coli* and *Klebsiella pneumoniae*? An in vivo study in *Galleria mellonella* with important implications for maximum residue limits in food)

Dear Prof. Christopher Kenyon:

Thank you for the privilege of reviewing your work. Below you will find my comments, instructions from the Spectrum editorial office, and the reviewer comments.

Revision Guidelines

Sincerely,
Sadjia Bekal
Editor
Microbiology Spectrum

Reviewer #1 (Comments for the Author):

The authors exposed *E. coli* and *K. pneumoniae* to ciprofloxacin in larvae of the moth *G. mellonella*. Exposure of a *K.p.* strain to ciprofloxacin at concentrations significantly below the MIC promoted mutations in genes increasing the MIC. The conclusions are that foodborne exposure to FQs could be within the range of MSCs, and that standards for foodborne MRLs should be

revisited accordingly. Reference to the mode of action of FQ [DNA gyrase], the SOS response, the decrease in DNA replication fidelity, and the accrual of mutations should be made in the discussion. The conclusions reported here may perhaps not be generalizable to antibiotics that don't set off the SOS response. Reference could be made to a recent paper speculating that this phenomenon will be of concern for humans: Subirats J, et al. Does dietary consumption of antibiotics by humans promote antibiotic resistance in the gut microbiome? Food Prot. 2019 Oct;82(10):1636-1642. doi: 10.4315/0362-028X.JFP-19-158. The paper could use some critical editing by the authors, it needs to be tightened up in a number of respects. Lines 95 and 97 would seem to contradict each other.

L. 107. Do you mean lower? As line 98 indicates an MSC of 4 ng/L, apparently in contradiction.

I. 207. This is not the concentration of cipro injected, rather the amount.

I. 216. Why kill dead larvae?

I. 290. What is the reference draft genome?

Fig. 1. A control consisting of uninoculated larvae is needed to confirm that the bacteria recovered are what was put in. Hopefully the authors have this in hand.

Table 2. The legend needs to describe what is in the table.

Fig. 4. The font size needs to be larger, as it is very hard to read. But, the utility of this figure is not clear. Suggest deleting it.

SFig5. Figure legend is very confusing.

Stab1 and 2. What does NA mean? #N/A?

The reference section needs to be uniformly formatted. Reference 16 appeared in 2017.

Reviewer #3 (Comments for the Author):

The authors hypothesized that the residual concentrations of antimicrobials allowed in the food we eat (Acceptable Daily Intake - ADIs) are able to select antimicrobial resistance in our resident microbiota. They found that the concentrations of ciprofloxacin/enrofloxacin allowed in food can induce de-novo ciprofloxacin resistance in *Klebsiella pneumoniae* but not in *E. coli*. The study is well designed and done. The data shown are interesting and worth to be published.

few commensts

- The author tried only ciprofloxacin/enrofloxacin. In that case please replace "antibiotics" for "fluoroquinolones". Also, why didn't try other classes of antibiotics ?

- Do you have explanation why the ciprofloxacin wasn't induced in *E. coli*?

- Did you try to check those low doses of antibiotics ingested could enhance plasmid transfer and, consequently, resistance gene exchange in the resident microbiota.

Reply to reviewers:

Reviewer #1 (Comments for the Author):

The authors exposed *E. coli* and *K. pneumoniae* to ciprofloxacin in larvae of the moth *G. mellonella*. Exposure of a *K.p.* strain to ciprofloxacin at concentrations significantly below the MIC promoted mutations in genes increasing the MIC. The conclusions are that foodborne exposure to FQs could be within the range of MSCs, and that standards for foodborne MRLs should be revisited accordingly. Reference to the mode of action of FQ [DNA gyrase], the SOS response, the decrease in DNA replication fidelity, and the accrual of mutations should be made in the discussion. The conclusions reported here may perhaps not be generalizable to antibiotics that don't set off the SOS response. Reference could be made to a recent paper speculating that this phenomenon will be of concern for humans: Subirats J, et al. Does dietary consumption of antibiotics by humans promote antibiotic resistance in the gut microbiome? *Food Prot.* 2019 Oct;82(10):1636-1642. doi: 10.4315/0362-028X.JFP-19-158. The paper could use some critical editing by the authors, it needs to be tightened up in a number of respects.

Lines 95 and 97 would seem to contradict each other.

Reply:

Thank you for the useful comments and suggestions. We have tried to tighten up the language and make sure we do not contradict ourselves anywhere. We have added the reference suggested by the reviewer. A new section has been added to the discussion to discuss the relevance of the SOS response (L437-444):

It would be useful if these studies could include other pathways through which low dose fluoroquinolones could select for AMR. Low dose fluoroquinolones have been found to activate the SOS response due to DNA damage caused by interference with DNA gyrase [1, 2]. This could lead to enhanced horizontal gene transfer via plasmids which could in turn induce AMR [1, 3]. In addition, ciprofloxacin has been shown to induce persister cells via the SOS response [2]. Other studies have found that these persister cells may be more likely to acquire AMR [4].

We have referred to how mutations in the QRDR of *gyrA* and *B* lead to fluoroquinolone resistance at lines 390-391.

L. 107. Do you mean lower? As line 98 indicates an MSC of 4 ng/L, apparently in contradiction.

Reply:

Thank you for pointing this error out. The wording has been changed to "lower".

L. 207. This is not the concentration of cipro injected, rather the amount.

Reply:

Agreed. We have changed this heading to: (L204)

Dose of ciprofloxacin injected

I. 216. Why kill dead larvae?

Reply:

This sentence has been edited to the following to address this issue: (L214-216)

When each experiment was completed, both the surviving and dead *G. mellonella*, were kept at -80°C overnight to sedate and kill the surviving larvae.

I. 290. What is the reference draft genome?

Reply:

The reference draft genomes are now included in lines 300-301.

The quality-controlled reads were mapped to respective reference draft genomes [M14827-2-A (STable1) and M17125-1A (STable2)] using CLC genomics Workbench (v20).

Fig. 1. A control consisting of uninoculated larvae is needed to confirm that the bacteria recovered are what was put in. Hopefully the authors have this in hand.

Reply:

We used three methods to ensure that the bacteria we injected were the bacteria we extracted. Firstly, we injected two groups of larvae ($n=10$, per group) with PBS and cultured these on selective *E. coli* and *K. pneumoniae* plates. We found that the haemolymph from these larvae did not yield any growth of *E. coli* or *K. pneumoniae* (L237-244 and 302-303). Secondly, we conducted whole genome sequencing of the *K. pneumoniae* from pre and post larval infection. The post -infection *K. pneumoniae* recovered was nearly identical genetically (belonged to the same ST type) to pre-infection isolate – with the notable exception of the resistance associated mutations. *K. pneumoniae* was the only species that grew on the antibiotic plates. Thirdly, we assessed if *E. coli* was recovered from the group of larvae that were infected with *K. pneumoniae* and vice versa. In both groups, only the inoculated *Enterobacteriales* species was recovered. We have also conducted a range of other infection experiments in *G. mellonella* over the past few years with a range of *Klebsiella*, *Neisseria*, *Enterococcus* and *Streptococcus* species [5-7]. We have not detected any *K. pneumoniae* in any of these experiments that we have not ourselves inoculated. In one of these experiments, where we injected *K. pneumoniae* M17125 into the larvae, we found via WGS that the same M17125 strain was the only strain of *K. pneumoniae* we retrieved from the haemolymph ([6]. Finally, *K. pneumoniae* has not been described in the *Galleria mellonella* microbiome [8].

Table 2. The legend needs to describe what is in the table.

Reply:

The legend and column headings have been edited to make this clearer.

Fig. 4. The font size needs to be larger, as it is very hard to read. But, the utility of this figure is not clear. Suggest deleting it.

Reply:

This figure has been deleted in the main manuscript and added to the online supplement.

SFig5. Figure legend is very confusing.

Reply:

This legend has been reworded to the following to enhance clarity:

SFigure 5. Ciprofloxacin MICs of bacterial colonies of *K. pneumoniae* M14827 after exposure to daily ciprofloxacin (ADI dose, 0.1xADI dose or control/PBS). Mean and SEM shown.

Stab1 and 2. What does NA mean? #N/A?

Reply:

N/A refers to not applicable or not detected. This definition has been added to the tables

The reference section needs to be uniformly formatted. Reference 16 appeared in 2017.

Reply:

The reference section has been reformatted, and the year of publication of reference 16 changed to 2017.

Reviewer #3 (Comments for the Author):

The authors hypothesized that the residual concentrations of antimicrobials allowed in the food we eat (Acceptable Daily Intake - ADIs) are able to select antimicrobial resistance in our resident microbiota. They found that the concentrations of ciprofloxacin/enrofloxacin allowed in food can induce de-novo ciprofloxacin resistance in *Klebsiella pneumoniae* but not in *E. coli*. The study is well designed and done. The data shown are interesting and worth to be published.

few commensts

- The author tried only ciprofloxacin/enrofloxacin. In that case please replace "antibiotics" for "fluoroquinolones". Also, why didn't try other classes of antibiotics ?

Reply:

The title has been changed to the following:

Could traces of fluoroquinolones in food induce ciprofloxacin resistance in *Escherichia coli* and *Klebsiella pneumoniae*? An *in vivo* study in *Galleria mellonella* with important implications for maximum residue limits

The text in the abstract and manuscript main text has also been changed in a similar fashion.

We are in the process of broadening the number of bug-drug combinations tested. We have finished a study evaluating the *in vivo* MSC_{denovo} of erythromycin in *S. pneumoniae* and are now evaluating tetracycline and trimethoprim MSCs in a range of species. We are also assessing *in vivo* MSC_{selects}.

- Do you have explanation why the ciprofloxacin wasn't induced in *E. coli*?

Reply:

We have expanded the section describing possible reasons for why we did not detect the emergence of ciprofloxacin resistance in the strain of *E. coli* we used in the discussion (lines 494-508):

Both *E. coli* and *K. pneumoniae* have been proposed as sensitive indicators of excess antimicrobial exposure in various environmental and clinical settings [9, 10]. For both species, there is now clear evidence that certain strains are more susceptible to the acquisition and spread of AMR [10]. For example, the ST131 H30-Rx strain of *E. coli* has been strongly linked to the spread of the ESBL enzyme, CTX-M15, as well as fluoroquinolone and β -lactam resistance [11]. In a similar vein, the clonal group 258 of *K. pneumoniae* has been shown to be responsible for most hospital-acquired infections of carbapenem-resistant *K. pneumoniae* [12]. In addition, *Klebsiella spp.* have been shown to play a critical role in the genesis and spread of AMR genes between environmental and human microbial populations [13]. The ATCC 25922 strain of *E. coli* we used belongs to the phylogenetic group A of *E. coli*, which has been found to be less prone to the emergence of AMR [14]. Future experiments may

consider using *E. coli* strains from the phylogroups such as B2, which are more associated with the emergence of AMR [14, 15].

Two additional references have been added to this section.

- Did you try to check those low doses of antibiotics ingested could enhance plasmid transfer and, consequently, resistance gene exchange in the resident microbiota.

Reply:

We unfortunately did not check if low doses of antimicrobials could enhance plasmid transfer. Other studies have however done this and we plan to include this in future studies [3].

References

1. Shun-Mei E, Zeng J-M, Yuan H, Lu Y, Cai R-X, Chen C. Sub-inhibitory concentrations of fluoroquinolones increase conjugation frequency. *Microbial pathogenesis*. 2018;114:57-62.
2. Dörr T, Lewis K, Vulić M. SOS response induces persistence to fluoroquinolones in *Escherichia coli*. *PLoS genetics*. 2009;5(12):e1000760.
3. Hallal Ferreira Raro O, Poirel L, Tocco M, Nordmann P. Impact of veterinary antibiotics on plasmid-encoded antibiotic resistance transfer. *Journal of antimicrobial Chemotherapy*. 2023;78(9):2209-16.
4. Levin-Reisman I, Ronin I, Gefen O, Braniss I, Shores N, Balaban NQ. Antibiotic tolerance facilitates the evolution of resistance. *Science*. 2017;355(6327):826-30. Epub 2017/02/12. doi: 10.1126/science.aaj2191. PubMed PMID: 28183996.
5. Hofkens N, Gestels Z, Abdellati S, De Baetselier I, Gabant P, Martin A, et al. Microbisporicin (NAI-107) protects *Galleria mellonella* from infection with *Neisseria gonorrhoeae*. *Microbiology Spectrum*. 2023:e02825-23.
6. Kenyon C, Gestels Z, Vanbaelen T, Britto B, Manoharan-Basil SS. Doxycycline PEP can induce doxycycline resistance in *Klebsiella pneumoniae* in a *Galleria mellonella* model of PEP. *Frontiers in Microbiology*. 2023;14:1208014.
7. Nele Hofkens, Zina Gestels, Said Abdellati, Irith De Baetselier, Philippe GABANT, Anandi MARTIN, et al. Microbisporicin (NAI-107) protects *Galleria mellonella* from infection with *Neisseria gonorrhoeae*. *Microbiology Spectrum*. In Press.
8. Allonsius CN, Van Beeck W, De Boeck I, Wittouck S, Lebeer S. The microbiome of the invertebrate model host *Galleria mellonella* is dominated by *Enterococcus*. *Animal Microbiome*. 2019;1:1-7.
9. Berendonk TU, Manaia CM, Merlin C, Fatta-Kassinos D, Cytryn E, Walsh F, et al. Tackling antibiotic resistance: the environmental framework. *Nat Rev Microbiol*. 2015;13(5):310-7. Epub 2015/03/31. doi: 10.1038/nrmicro3439. PubMed PMID: 25817583.
10. Andersson DI, Balaban NQ, Baquero F, Courvalin P, Glaser P, Gophna U, et al. Antibiotic resistance: turning evolutionary principles into clinical reality. *FEMS Microbiology Reviews*. 2020;44(2):171-88.

11. Price LB, Johnson JR, Aziz M, Clabots C, Johnston B, Tchesnokova V, et al. The epidemic of extended-spectrum- β -lactamase-producing *Escherichia coli* ST131 is driven by a single highly pathogenic subclone, H 30-Rx. *MBio*. 2013;4(6):e00377-13.
12. Wyres KL, Holt KE. *Klebsiella pneumoniae* population genomics and antimicrobial-resistant clones. *Trends in microbiology*. 2016;24(12):944-56.
13. Wyres KL, Holt KE. *Klebsiella pneumoniae* as a key trafficker of drug resistance genes from environmental to clinically important bacteria. *Current opinion in microbiology*. 2018;45:131-9.
14. Rezaatofghi SE, Najafifar A, Askari Badouei M, Peighambari SM, Soltani M. An integrated perspective on virulence-associated genes (VAGs), antimicrobial resistance (AMR), and phylogenetic clusters of pathogenic and non-pathogenic avian *Escherichia coli*. *Frontiers in veterinary science*. 2021;8:758124.
15. Wang X-M, Jiang H-X, Liao X-P, Liu J-H, Zhang W-J, Zhang H, et al. Antimicrobial resistance, virulence genes, and phylogenetic background in *Escherichia coli* isolates from diseased pigs. *FEMS microbiology letters*. 2010;306(1):15-21.

Re: Spectrum03595-23R1 (Could traces of fluoroquinolones in food induce ciprofloxacin resistance in Escherichia coli and Klebsiella pneumoniae? An in vivo study in Galleria mellonella with important implications for maximum residue limits in food)

Dear Prof. Christopher Kenyon:

Your manuscript has been accepted, and I am forwarding it to the ASM production staff for publication. Your paper will first be checked to make sure all elements meet the technical requirements. ASM staff will contact you if anything needs to be revised before copyediting and production can begin. Otherwise, you will be notified when your proofs are ready to be viewed.

Sincerely,
Sadjia Bekal
Editor
Microbiology Spectrum